

# A retrospective analysis of postoperative hypokalemia in pituitary adenomas after transsphenoidal surgery

Lili You[1,*], Wenpeng Li[2,*], Tang Chen[2], Dongfang Tang[2], Jinliang You[2] and Xianfeng Zhang[2]

[1] Department of Clinical Epidemiology, First Hospital of Jilin University, Changchun, China
[2] Department of Neurosurgery, First Hospital of Jilin University, Changchun, China
[*] These authors contributed equally to this work.

## ABSTRACT

**Background**. Pituitary adenoma is one of the most common intracranial neoplasms, and its primary treatment is endoscopic endonasal transsphenoidal tumorectomy. Postoperative hypokalemia in these patients is a common complication, and is associated with morbidity and mortality. This study aimed to analyze the etiopathology of postoperative hypokalemia in pituitary adenomas after endoscopic transsphenoidal surgery.

**Methods and Materials**. This retrospective study included 181 pituitary adenomas confirmed by histopathology. Unconditional logistic regression analysis was used to calculate odds ratios (ORs) and 95% confidence intervals (CIs). Repeated measures ANOVA was used to analyze change in serum potassium levels at different time points.

**Results**. Multiple Logistic regression analysis revealed that only ACTH-pituitary adenoma (OR = 4.92, 95% CI [1.18–20.48], $P = 0.029$) had a significant association with postoperative hypokalemia. Moreover, the overall mean serum potassium concentration was significantly lower in the ACTH versus the non-ACTH group (3.34 mmol/L vs. 3.79 mmol/L, $P = 0.001$). Postoperative hypokalemia was predominantly found in patients with ACTH-pituitary adenoma ($P = 0.033$).

**Conclusions**. ACTH-pituitary adenomas may be an independent factor related postoperative hypokalemia in patients despite conventional potassium supplementation in the immediate postoperative period.

## INTRODUCTION

Pituitary adenomas account for approximately 15% of all intracranial neoplasms (*Ezzat et al., 2004*; *Gold, 1981*) and have characteristic clinical manifestations due to overproduction or insufficient secretion of hypophyseal hormones and/or local mass effects (*Mete & Asa, 2012*; *Scangas & Laws, 2014*). Pituitary adenomas are categorized by secretory activity; null cell adenomas and prolactinomas (PRL) are the most common pituitary adenomas, followed by growth hormone (GH)-pituitary adenomas, adrenocorticotropic hormone (ACTH)-pituitary adenomas, follicle-stimulating hormone (FSH)-pituitary adenomas, and

Corresponding author
Xianfeng Zhang,
18343113238@163.com

thyroid-stimulating hormone (TSH)-pituitary adenomas (*Ezzat et al., 2004*; *Lake, Krook & Cruz, 2013*; *Melmed, 2015*). Pituitary adenomas are usually monoclonal benign epithelial tumors and rarely turn malignant (*Melmed, 2011*; *Melmed, 2015*). Endonasal transsphenoidal surgery has proved a safe and efficacious treatment of pituitary adenomas (*Constantino et al., 2016*; *Wang et al., 2015*; *Zhan et al., 2015*).

Hypokalemia is defined as a value of serum potassium concentration less than 3.5 mmol/L, and is the most common electrolyte abnormality encountered in clinical practice (*Glover, 1999*; *Halperin & Kamel, 1998*). Most patients have mild hypokalemia, but nearly one quarter have serum potassium concentrations below 2.5 mmol/L, defined as severe hypokalemia, which can cause many signs and symptoms such as fatigue, nausea, vomiting, and muscle weakness (*Lodin & Palmer, 2015*; *Weir & Espaillat, 2015*). Moreover, unrecognized hypokalemia can lead to respiratory failure and is associated with morbidity and mortality (*Wojtaszek & Matuszkiewicz-Rowinska, 2013*).

This study aimed to investigate the factors that influence the outcome of postoperative hypokalemia in a consecutive series of patients with pituitary adenomas after transsphenoidal surgery.

## MATERIALS AND METHODS

### Study population
We retrospectively reviewed the medical records of 181 consecutive patients of pituitary adenomas from the Department of Neurosurgery, First Hospital of Jilin University (Changchun, China) treated from January 2010 to December 2012. All patients underwent endoscopic endonasal transsphenoidal surgery, and tumor resection was carried out by the same surgeon. Among these, 115 patients had histopathology-proven functioning adenomas and 66 had clinically nonfunctioning adenomas. We then analyzed data obtained from all cases. None of the subjects received chemotherapy or radiotherapy before/after surgery, and patients with previous pituitary surgery and lesions other than pituitary adenomas were excluded from this study. Further, all cases included in this study accepted total tumorectomy, and cases that underwent selective adenectomy were excluded. All patients were of Han descent from the Changchun area. Patient characteristics and clinical details were obtained from medical records. We adhered to the bioethics principles of the Declaration of Helsinki, and this study was approved by the Ethics Committee of the First Hospital of Jilin University (Reference Number: 2016-324).

### Preoperative evaluation
All patients mainly underwent a preoperative neuroradiological and biochemical evaluation. Additionally, baseline information such as sex, age at operation, type of residence, rural/urban geography, length of stay, and history of hypertension or diabetes were included.

All patients underwent diagnostic computed tomographic (CT) scanning, and some had magnetic resonance imaging (MRI) in the sellar region to further determine tumor type. Adenoma dimensions were recorded from neuroradiological images, and tumor was classified according to size based on its maximum diameter into two categories: microadenoma ($<1.0$ cm) and macroadenoma ($\geq 1.0$ cm).

**Table 1  Diagnostic tests useful in the evaluation of the suspected pituitary adenomas.**

| Tumor type | Diagnostic tests | Reference range | Diagnostic tests notes |
|---|---|---|---|
| PRL-pituitary adenoma | Serum prolactin | Elevated ($\geq$250 mcg/L ) | 25–249 mcg/L should prompt investigation of other causes of hyperprolactinemia |
| GH-pituitary adenoma | Insulinlike growth factor 1 | Elevated (76–328 ng/mL) | Normally elevated during pregnancy |
| | Oral glucose suppression test | Elevated (0–1 ng/mL) | Failure of growth hormone to decrease to <1 ng/mL two hours after administering 75 g of oral glucose |
| ACTH-pituitary adenoma | 24-h urine free cortisol | Elevated (10–84 mcg total/24-h period) | Total high false-positive rate in women taking estrogen diagnostic if four times greater than normal |
| | Late-night salivary cortisol | Elevated (0.01–0.09 mcg/dL) | Midnight sample |
| | 1-mg overnight dexamethasone suppression | Elevated (cortisol $\geq$ 1.8 ng/dL) | High false-positive rate in women taking estrogen and further testing needed to rule out the source of excess cortisol and to rule out "pseudo–Cushing syndrome" |
| FSH-pituitary adenoma | FSH | FSH (2–35 mIU/mL) | In postmenopausal women, elevated FSH levels are normal, and value for menstruating women varies based on phase of menstrual cycle |
| TSH-pituitary adenoma | TSH | Elevated (0.5–4.8 mIU/L) | May be atypically normal in relation to free T4 |
| | Free T4 | Low (4.2–13 ng/dL) | Low T4 with normal or low TSH indicates secondary hypothyroidism (possibly from pituitary dysfunction) |
| Mixed- pituitary adenoma | No specific corresponding test | Combination of hormones | Varies based on dominant hormone |
| Nonfunctioning-adenoma | None | None | None |

Biochemical examination should carefully screen the hypothalamus–pituitary–adrenal (HPA) axis, with a focus on pituitary function, to check for preoperative endocrine excess or insufficiency. Multiple measurements of plasma PRL, GH, ACTH, FSH, and TSH were done. In addition, levels of free thyroxine, insulin-like factor-1 (IGH-1), and 24-h urinary free cortisol as well as glucose on an oral glucose tolerance test (OGTT) were measured when necessary, and the details of the endocrine disgnosis for pituitary adenomas in each types are shown in Table 1. Secretory syndromes must be excluded when pituitary adenomas are preoperatively diagnosed; therefore, a diagnosis of pituitary adenoma must be based on combined results from imaging and endocrinology.

## Postoperative evaluation

All subjects in this study consented to endoscopic endonasal transsphenoidal surgery. Histopathologic examination was conducted immediately at the pathology department of the same hospital for a confirmative diagnosis of adenoma type. Four micrometers thick serial sections of the paraffin-embedded block was excised, and sections from each case were stained with routine Hematoxylin and Eosin method for histopathologic evaluation. Paraffin sections of each tumor were immunostained using the primary antibodies against the following pituitary hormones: PRL, GH, ACTH, FSH, TSH, LH (Zhongshan, Beijing, China). Visualization of the immune reactions was done by Streptavidin–Biotin-peroxidase

technique, and 3, 3-diaminobenzidine was employed as a chromogen. The immunostaining results for each patient were graded as being 0 (negative), 1 + (10–30% of cells), 2 + (30–50% of cells) or 3 + (over 50% of cells) by the pathologists. The presence of more than 10% of hormone immunopositive cells was considered secretory tumor. The tumors with high co-expression immunoreactivity were considered Mix-pituitary adenomas.

In the early postoperative period, patients were treated in the intensive care unit (ICU) and, at the same time, a CT scan was done to check surgical outcome. Levels of serum ions were measured and a neurological examination and visual field assessment were conducted and recorded for all patients with pituitary adenoma. Follow-up MRI was performed on the first postoperative day and, thereafter, at three and six months.

Levels of serum potassium were immediately measured postoperatively and repeated a total of four times until the third postoperative day for all patients who underwent surgery, and conventional potassium supplementation was done to prevent postoperative hypokalemia. For patients with confirmed hypokalemia, serum potassium was monitored until concentrations returned to normal. Intravenous potassium supplementation was initiated for patients with intractable hypokalemia. In general, serum potassium levels would become normal within 2–3 days.

To explore the relationship between ACTH-pituitary adenomas and postoperative hypokalemia, 16 cases of hypokalemia patients were included, of which eight can be classified as ACTH group and the others were non-ACTH group. Cases would be included in the ACTH group if: (1) the clinical manifestation of patients were mainly including central obesity, hypertension, facial plethora, proximal muscle weakness, dacreased libido or impotence and so on; (2) patients underwent CT scanning or MRI examination confirmed the presence of intracranial tumors; and (3) biochemical examination results of 24-h urine free cortisol, late-night salivary cortisol and 1-mg overnight dexamethasone suppression were positive (levels of results were elevated). If not, the patients would been divided into non-ACTH group.

## Criteria for hypokalemia

For patients with pituitary adenomas, parameters of hepatic and renal function, blood electrolytes, lipids and blood glucose were measured at admission, with a predominant focus on serum potassium, especially on the day of surgery. Data on serum potassium were recorded by obtaining a daily list of medical inpatients from the database of the biochemistry laboratory, where serum potassium was measured by direct ion selective method using an autoanalyzer (DXC800-(5416); Beckman Coulter, Brea, CA, USA). Normal serum potassium levels range from 3.5 to 5.5 mmol/L, and a level <3.5 mmol/L is defined as hypokalemia. The patient's medical records were located and evaluated to identify if they had mild, moderate, or severe hypokalemia, in the range of 3.0–3.5 mmol/L, 2.5–3.0 mmol/L, and <2.5 mmol/L levels, respectively.

## Statistical analysis

Demographics and baseline characteristics of subjects were presented as mean values ± standard deviation for continuous data with normal distribution. Categorical data

were displayed as percentage and frequency. Continuous and normal distribution data were compared by Student's $t$ test or ANOVA. Categorical data were compared using the chi-square or Fisher's exact test, as appropriate. Repeated measures ANOVA was used to analyze changes in serum potassium levels with time, Mauchly's test of sphericity should be used to judge whether there were relations among the repeatedly measured datas. If any ($P < 0.05$), multivariate ANOVA should be taken next, or Greenhouse-Geisser corrected results should be taken. Treated effect could be evaluated by estimating between the subject variance. Repeated measurement effect or its interactive effect with treated group could be evaluated by estimating within subject variance. Further, unconditional logistic regression analysis was used to calculate odds ratios (ORs) and 95% confidence intervals (CIs). For independent predictive factors of hypokalemia, multivariate logistic regression analysis was performed, with a significance level of $P < 0.05$ for inclusion and $P > 0.10$ for exclusion of variables and the stepwise selection method was used to automatically choose the variables. All statistical assessments were performed using SPSS18.0 software (SPSS Inc., Chicago, IL, USA). Two-tailed $P$-values <0.05 were considered indicative of statistical significance.

# RESULTS

## Subject characteristics

In total, 181 pituitary adenomas (patient mean age 46.6 ± 13.2 years at operation; 76 male (42.0%; mean age 48.5; range 17–81 years) and 105 female (58.0%; mean age 45.8; range 14–68 years)) underwent endoscopic endonasal transsphenoidal resection. All patients were treated at a single medical center and by the same surgeon (Dr Gang Zhao, Director of the Department of Neurosurgery, The First Hospital of Jilin University). Of the 181 subjects included in the study, 66 (36.5%) had a nonfunctioning pituitary adenoma, 49 (27.1%) had PRL-pituitary adenomas, 28 (15.5%) had GH-pituitary adenomas, 16 (8.8%) had ACTH-pituitary adenomas, nine (5.0%) had FSH-pituitary adenomas, and five (2.8%) had TSH-pituitary adenomas; moreover, eight (4.4%) mixed pituitary adenomas were included. By tumor size, there were only 30 (16.6%) microadenomas and the remaining 151 (83.4%) were macroadenomas. Patient characteristics are shown in Table 2.

## Association of clinical characteristics with risk of postoperative hypokalemia

To explore statistical association between demographic and clinical characteristics and postoperative hypokalemia, a multivariate logistic regression model was used. Only the ACTH-pituitary adenoma was associated with a statistically significant increased risk of postoperative hypokalemia compared to non-ACTH-pituitary adenomas (OR = 4.92; 95% CI [1.18–20.48], $P = 0.029$; Table 3). Univariate analyses of characteristics associated with the risk of postoperative hypokalemia are presented in Table 2. Rural/urban geography significantly contributed to the definition of the influence factors. Rural patients were more likely to have postoperative hypokalemia (OR = 3.89; 95% CI [1.07–14.14], $P = 0.040$). The OR for association of postoperative hypokalemia with length of hospital stay showed postoperative hypokalemia occurred more frequently in the group with stay ≥12 days (OR = 3.24; 95% CI [1.12–9.38], $P = 0.030$) than the other group (length of hospital stay

**Table 2 Postoperative hypokalemia by demographic and clinical datas: univariate analysis.**

| Variables | Patients (%) | Hypokalemia (%) | OR (95% CI) | P value |
|---|---|---|---|---|
| Sex | | | | |
| Male | 76 (42.0) | 7 (9.2) | 1.00 | 0.881 |
| Female | 105 (58.0) | 9 (8.6) | 0.92 (0.33–2.60) | |
| Age | | | | |
| ≤45 | 80 (44.2) | 6 (7.5) | 1.00 | 0.573 |
| >45 | 101 (55.8) | 10 (9.9) | 1.36 (0.47–3.90) | |
| Rural/urban geography | | | | |
| Urban | 81 (44.8) | 3 (3.7) | 1.00 | 0.040 |
| Rural | 100 (55.2) | 13 (13.0) | 3.89 (1.07–14.14) | |
| Length of stay | | | | |
| ≤12 days | 115 (63.5) | 6 (5.2) | 1.00 | 0.030 |
| >12 days | 66 (36.5) | 10 (15.2) | 3.24 (1.12–9.38) | |
| Hypertension | | | | |
| No | 146 (80.7) | 8 (5.5) | 1.00 | 0.003 |
| Yes | 35 (19.3) | 8 (22.9) | 5.11 (1.76–14.80) | |
| Diabetes | | | | |
| No | 162 (89.5) | 12 (7.4) | 1.00 | 0.059 |
| Yes | 19 (10.5) | 4 (21.1) | 3.33 (0.96–11.63) | |
| Tumor size | | | | |
| ≤1.0 cm | 30 (16.6) | 2 (6.7) | 1.00 | 0.648 |
| >1.0cm | 151 (83.4) | 14 (9.3) | 1.43 (0.31–6.65) | |
| Type | | | | |
| Nonfunctioning | 66 (36.5) | 2 (3.0) | 1.00 | 0.054 |
| Functioning | 115 (63.5) | 14 (12.2) | 4.44 (0.98–20.17) | |
| Type | | | | |
| Non-PRL | 132 (72.9) | 14 (10.6) | 1.00 | 0.186 |
| PRL | 49 (27.1) | 2 (4.1) | 0.36 (0.08–1.64) | |
| Type | | | | |
| Non-GH | 153 (84.5) | 14 (9.2) | 1.00 | 0.732 |
| GH | 28 (15.5) | 2 (7.1) | 0.76 (0.16–3.56) | |
| Type | | | | |
| Non-ACTH | 165 (91.2) | 8 (4.8) | 1.00 | <0.001 |
| ACTH | 16 (8.8) | 8 (50.0) | 19.63 (5.82–65.84) | |
| Type | | | | |
| Non-FSH | 172 (95.0) | 16 (9.3) | 1.00 | 0.722 |
| FSH | 9 (5.0) | 0 | – | |
| Type | | | | |
| Non-TSH | 176 (97.2) | 15 (8.5) | 1.00 | 0.391 |
| TSH | 5 (2.8) | 1 (6.3) | 2.68 (0.28–25.57) | |
| Type | | | | |
| Non-Mix | 173 (95.6) | 15 (8.7) | 1.00 | 0.711 |
| Mix | 8 (4.4) | 1 (12.5) | 1.51 (0.17–13.06) | |

**Table 3  Postoperative hypokalemia by demographic and clinical datas: multivariate analysis.**

| Variables | β | OR (95% CI) | P |
|---|---|---|---|
| Geography (Rural vs. Urban) | 1.01 | 2.74 (0.69–10.88) | 0.151 |
| Length of stay (>12 days vs. ≤12 days) | 0.96 | 2.62 (0.81–8.51) | 0.109 |
| Hypertension (Yes vs. No) | 1.21 | 3.35 (0.93–12.09) | 0.065 |
| ACTH-pituitary adenoma (Yes vs. No) | 1.59 | 4.92 (1.18–20.48) | 0.029 |

<12 days). In comparison with non-hypertensive patients, patients with hypertension had a higher OR of postoperative hypokalemia (OR = 5.11; 95% CI [1.76–14.80], $P = 0.003$). ACTH-pituitary adenomas were associated with an increased OR of postoperative hypokalemia compared with non-ACTH-pituitary adenomas (OR = 19.63; 95% CI [5.82–65.84], $P < 0.001$), whereas no statistically significant differences in incidence rates between postoperative hypokalemia and the remaining factors (Table 2) were evident.

## Variations of serum potassium at different time points in postoperative hypokalemia

Of the 181 subjects with pituitary adenoma, 16 patients had postoperative hypokalemia. The group with patients who developed postoperative hypokalemia comprised eight ACTH, two PRL, two non-functioning, two GH, one TSH, and 1 mixed pituitary adenoma. To evaluate the impact of ACTH on postoperative hypokalemia in patients with pituitary adenomas, we subdivided the study sample into the ACTH-pituitary adenoma group (ACTH group) and the non-ACTH-pituitary adenoma group (control group) and compared serum potassium levels between the two groups. From the third postoperative day onwards, the overall mean serum potassium concentration was significantly lower in the ACTH group than in the control group (3.34 mmol/L vs. 3.79 mmol/L, $P = 0.001$).

The serum potassium concentration was measured and recorded 4 times until the third postoperative day. On the day of surgery, mean serum potassium in the ACTH and control groups was 2.92 and 3.32 mmol/L, respectively ($P = 0.035$); on the first postoperative day, the mean serum potassium concentration was significantly lower in the ACTH group compared to control group (ACTH 3.14 mmol/L; control 3.71 mmol/L, $P = 0.021$). On the second and third postoperative days, the mean serum potassium was 3.46 and 3.86 mmol/L in the ACTH group, and 3.94 and 4.21 mmol/L in the control group, without any statistically significant between-group differences ($P = 0.058$ and $P = 0.150$, respectively; Table 4). Levels of serum potassium in the study groups at each time point are shown in Fig. 1. Summary statistics for average serum potassium and levels at the four time points they were measures at are presented in Table 4. Repeated measures ANOVA was used to analyze the interaction between the subgroup of postoperative hypokalemia and the measurement time points, the result showed that the interaction between postoperative hypokalemia and time is not statistically significant ($P = 0.321$), and a statistically significant correlation and that postoperative hypokalemia was predominant among patients with ACTH-pituitary adenomas ($P = 0.033$).

**Table 4  Impacts of different groups on postoperative hypokalemia.**

| Group | | Measured at different time | | | | Sum | F | P |
|---|---|---|---|---|---|---|---|---|
| | | T1 | T2 | T3 | T4 | | | |
| ACTH group | $\overline{x}$ | 2.92 | 3.14 | 3.46 | 3.86 | 3.34 | 4.552 | 0.010 |
| | s | 0.43 | 0.54 | 0.58 | 0.60 | 0.63 | | |
| Control group | $\overline{x}$ | 3.32 | 3.71 | 3.94 | 4.21 | 3.79 | 28.356 | <0.001 |
| | s | 0.09 | 0.20 | 0.21 | 0.26 | 0.38 | | |
| Total | $\overline{x}$ | 3.12 | 3.43 | 3.70 | 4.03 | 3.57 | 5.625[a] | 0.033[a] |
| | s | 0.36 | 0.49 | 0.49 | 0.48 | 0.56 | | |
| t | | 2.567 | 2.792 | 2.186 | 1.522 | $(F = 1.166, P = 0.321)$[b] | | |
| P | | 0.035 | 0.021 | 0.058 | 0.150 | | | |

**Notes.**

[a] $F$ statistic and $P$ value of main effect.

[b] $F$ statistic and $P$ value of crossover effect, and the mean of crossover effect is the interaction between postoperative hypokalemia and time.

T1, day of surgery; T2, postoperative day 1; T3, postoperative day 2; T4, postoperative day 3.

# DISCUSSION

The ACTH-secreting pituitary adenoma is related to a clinical disorder known as Cushing's disease (CD), which causes adrenal hypercortisolemia and consequent osteoporosis, muscle atrophy, psychiatric disorders, and, ultimately, death (*Castinetti et al., 2012*; *Sulentic, Morris & Grossman, 2000*). ACTH-pituitary adenomas are recognized as a more aggressive and invasive subtype of pituitary adenomas (*Jesser, Schlamp & Bendszus, 2014*; *Lake, Krook & Cruz, 2013*; *Maragliano et al., 2015*). Total-body potassium is predominantly present as an intracellular component. In clinical medicine, hypokalemia is one of the most frequently electrolyte abnormalities, affects multiple organ systems, and contributes to a significant risk of morbidity and mortality (*Crop et al., 2007*; *Jordan & Caesar, 2015*).

In all of the 181 subjects in this study, serum potassium levels were normal preoperatively. However, 16 (8.8%) patients with pituitary adenomas developed postoperative hypokalemia although conventional potassium supplementation was instituted immediately after surgery. This electrolyte disturbance is neither drug nor management related, and appears during the intraoperative period. Serum potassium was lower in the ACTH group compared to the control group, indicating that ACTH may play an important role in postoperative hypokalemia. Besides, compared with non-ACTH pituitary adenomas, the levels of serum potassium in ACTH pituitary adenomas patients were lower in the day and the first day of postoperation, but this phenomenon disappeared in the second day and the third day of postoperation. According to these results, ACTH-pituitary adenomas was significant association with the elevated incidence of hypokalemia and reducing the ability of patients with hypokalemia to recover normal revels of potassium.

It has been demonstrated that CD can cause hypokalemia because of the changing levels of hormones (*Campusano et al., 1999*; *Fernandez-Rodriguez et al., 2008*). Therefore, potassium supplementation is undertaken for patients with ACTH-pituitary adenoma. Preoperative surgical planning includes confirmation that serum potassium levels are normal. Moreover, conventional potassium supplementation has been carried out in the immediate

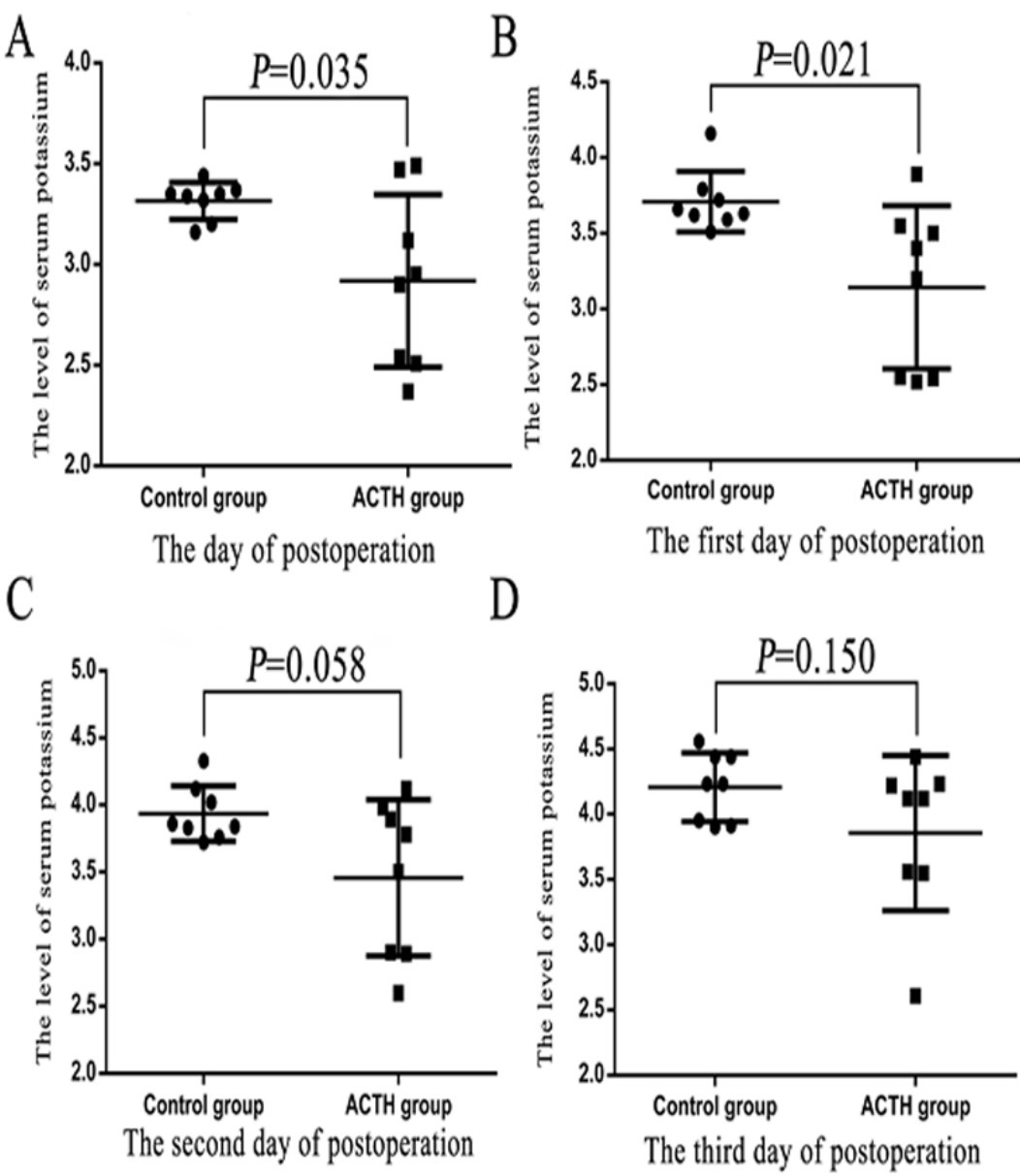

**Figure 1  The Student's *t* test was used to analyze the levels of serum potassium after the operation of patients with postoperative hypokalemia in ACTH group and control group.** (A) The day of postoperation. (B) The first day of postoperation. (C) The second day of postoperation. (D) The third day of postoperation.

postoperative period as a preventive measure. In this study, however, half of the patients with postoperative hypokalemia were those with ACTH-pituitary adenomas who were pre-operatively drug-naïve of potassium supplementation. We hypothesized intraoperative low potassium content was the cause of postoperative hypokalemia. First, it is well known that glucocorticoid and mineralocorticoid release intraoperatively usually counters the fall in serum potassium (*Chauhan et al., 2015*; *Fox et al., 2016*; *Frindt & Palmer, 2012*; *Lang &*

*Vallon, 2012*; *Ohtake et al., 2014*; *Salyer et al., 2013*; *Terker & Ellison, 2015*). Moreover, patients with ACTH-pituitary adenomas are predisposed to high glucocorticoid levels, which can cause hypokalemia (*Bondugulapati et al., 2016*; *Carrasco & Villanueva, 2014*; *Cassarino et al., 2017*). Second, all subjects in this study underwent endoscopic endonasal transsphenoidal tumorectomy, wherein cells of the pituitary gland were destroyed thus releasing hormones intraoperatively; simultaneously, tumor tissues were extruded and further hormone secretion. In the group with ACTH-pituitary adenomas, ACTH levels increased greatly in a short time. To further confirm the effect of surgical resection on postoperative hypokalemia, we should intraoperatively monitor changes in ACTH and serum potassium levels. This study elucidates the potential etiopathology of postoperative hypokalemia in patients with pituitary adenomas.

There are certain limitations of our study that should be acknowledged. First, only 16 patients with ACTH-pituitary adenomas were included in our study. A moderate sample size prevented assessment of the effects of the clinical characteristics of ACTH-pituitary adenomas on postoperative hypokalemia. Second, we only included subjects of ethnic Han lineage in northeastern China, and this may further introduce a heterogeneity with regard to the rest of the Han population in other regions. Further studies including a larger sample of patients with ACTH-pituitary adenomas are needed to validate our findings. Finally, inherent to the study's retrospective design, selection and information biases could not be excluded. In addition, the data from medical records and the retrospective nature of the case-control methodology represent limitations of this study because they preclude determining the causal direction of the variables analyzed with any certainty. Our elucidation of the causative pathology of postoperative hypokalemia in ACTH-pituitary adenomas was based on hypotheses inferred from our study results and should be further verified.

## CONCLUSIONS

In summary, ACTH-pituitary adenomas may cause postoperative hypokalemia in patients despite conventional potassium supplementation in the immediate postoperative period. However, more experimental research and clinical studies are needed to determine the influence of the ACTH-pituitary adenoma on postoperative hypokalemia and its etiopathologic mechanism.

### Funding
The authors received no funding for this work.

### Competing Interests
The authors declare there are no competing interests.
## Author Contributions

- Lili You, Wenpeng Li and Xianfeng Zhang conceived and designed the experiments, performed the experiments, analyzed the data, contributed reagents/materials/analysis tools, wrote the paper, prepared figures and/or tables, reviewed drafts of the paper.
- Tang Chen, Dongfang Tang and Jinliang You performed the experiments, contributed reagents/materials/analysis tools, reviewed drafts of the paper.

## Human Ethics

The following information was supplied relating to ethical approvals (i.e., approving body and any reference numbers):

This study was approved by the ethics committee of the First Hospital of Jilin University. We adhered to the bioethics principles of the Declaration of Helsinki, and our study was authorized by the Ethics Committee of the First Hospital of Jilin University (Reference Number: 2016-324).

## Data Availability

The raw data has been supplied as a Supplementary File.

## Supplemental Information

Supplemental information for this article can be found online at http://dx.doi.org/10.7717/peerj.3337#supplemental-information.

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
