# Peer review of "A retrospective analysis of postoperative hypokalemia in pituitary adenomas after transsphenoidal surgery"

_PeerJ, doi:10.7717/peerj.3337_

## Round 0.1 · original submission · Major Revisions

The authors tried to describe the etiopathological mechanism of the ACTH-pituitary adenoma with postoperative hypokalemia.
The authors have to specify the ACTH and non-ACTH pituitary adenomas groups of patients in material and methods. In the same section, some techniques are missing related to: histopathological examination; biochemical evaluations and hypokalemia analysis.
I agree with the authors when they say that there are 2 limitations in this study (a small sample size and the origin of the population). The authors should better specify that the etiopathological mechanism described is an hypothesis with poor supporting data, but it is a beginning for further studies.

·

Basic reporting

I have reviewed this paper from a statistical and methodological standopint.
No comment about: English, literature, article structure

Experimental design

Good experimental design; good use of statistical methods.
I only suggest the following minimal corrections:

1. line 112: the sentence should be modified in: "for continuous data with non-normal distribution"

2. line 113: the sentence should be modified in: "...exact test, as appropriate"

3. figure 1: the statistical test used to analyze data should be reported in figure legend (not only into the statistical analysis section)

No other comments about the experimental design

Validity of the findings

Authors fill a gap in scientific literature.

Additional comments

The paper is well-written and methodologically well-designed.
I only suggest minimal correction reported above.

Reviewer 2 ·

Basic reporting

Tables and figures need to be improved.

Experimental design

The research question needs to be re-defined. See my general comments to authors

Validity of the findings

Causal assertions are not justified in restrospective studies. Please revise the whole text in agreement with the study design. See my general comments to authors

Additional comments

The manuscript presents the results of a retrospective study including 181 pituitary adenomas which was aimed to investigate i) patient’s and pituitary adenoma’s characteristics which are associated with postoperative hypokalemia and ii) the difference of serum potassium levels between ACTH and non-ACTH pituitary adenomas. The manuscript is generally well-written, using clear and unambiguous English language, raw data are supplied and well-detailed, data is robust and statistically sound. However, in my opinion the research question needs to be completely re-defined because it appears too ambitious related to the retrospective study design, which is used. In fact, the authors in the abstract and the whole text affirm their paper is aimed to investigate the etiopathology of postoperative hypokalemia, but such a scope can be addressed only through prospective studies. Also their conclusion that “ACTH-pituitary adenomas may be an independent factor causing postoperative hypokalemia…” is inappropriate for retrospective studies, which are only prone to assess associations but not causal relations. In the following, some suggestions to improve the manuscript:
Abstract
Rewrite Background and Conclusions as suggested in my general comments.
lines 23-25: it is not clear what the authors mean with “...dependent variables which cause postoperative hypokalemia…” and “independent factor” used for ACTH-pituitary adenoma.
Material and Methods: Preoperative evaluation
lines 82-83: clarify and add references
Material and Methods: Statistical analysis
lines 111-112: It doesn’t seem to me that the authors have used continuous variables apart from serum potassium levels. In fact, age, LoS, tumor size have been categorized. Therefore, these lines concerning statistical tests not used can be deleted.
line 113: the authors should include the statistical test for sphericity and motivate the choice of the Greenhouse–Geisser test
line 117: the authors should explain if they used the stepwise selection method to automatically choose the variables
Results
line 129: the authors should include data for mixed pituitary adenomas in Table 1
line 134: what is the dependent variable? Clarify
lines 136,138: the acronym LOS can be deleted because not used anywhere
line 143-144: it is incorrect to use the term incidence for retrospective studies. The authors should write “to explore statistical association between demographic and clinical characteristics and postoperative hypokalemia”
lines 149-151: to move this sentence at the line 134. In fact, it is better for sake of clarity to present information about pituitary adenoma and postoperative hypokalemia first of all
lines 168: the interaction between postoperative hypokalemia and time is not statistically significant (p=0.321) (See Table 1)
Discussions
Rewrite this section as suggested in my general comments, avoiding causal assertions. The authors should discuss their finding also considering that potassium levels were significantly different between ACTH and non-ACTH only till the postoperative day 1, but that such a difference disappears from the 2nd postoperative day on.
lines 182-184: do not generalize relations observed only at univariate statistical analysis, but not confirmed at multivariate analysis
Table 1
Get the title more informative as, for example, “Postoperative hypokalemia by demographic and clinical characteristics: Univariate analysis”
Among the Variables, insert a variable “Type” with the following categories: ACTH vs non-ACTH, the latter sub-classified as PRL, GH, FSH, TSH, mixed.
Table 2
Get the title more informative as, for example, “Postoperative hypokalemia by demographic and clinical characteristics: Multivariate analysis”
Replace the word “Sum” with “Total”; explain the meaning of crossover effect in the note
Figure
the quality can be improved
Typos: lines 234, 250

---

## Round 0.2 · accepted · Accept

The authors of the paper completed the revision appropriately and improved the quality of the paper.